# Human iPSC-Derived Retinal Organoids and Retinal Pigment Epithelium for Novel Intronic *RPGR* Variant Assessment for Therapy Suitability

**DOI:** 10.3390/jpm12030502

**Published:** 2022-03-21

**Authors:** Fidelle Chahine Karam, To Ha Loi, Alan Ma, Benjamin M. Nash, John R. Grigg, Darshan Parekh, Lisa G. Riley, Elizabeth Farnsworth, Bruce Bennetts, Anai Gonzalez-Cordero, Robyn V. Jamieson

**Affiliations:** 1Eye Genetics Research Unit, Children’s Medical Research Institute, Sydney Children’s Hospitals Network, Save Sight Institute, University of Sydney, Westmead, Sydney 2145, Australia; fkaram@cmri.org.au (F.C.K.); tloi@cmri.org.au (T.H.L.); alan.ma@health.nsw.gov.au (A.M.); benjamin.nash@health.nsw.gov.au (B.M.N.); john.grigg@sydney.edu.au (J.R.G.); 2Department of Clinical Genetics, Western Sydney Genetics Program, Sydney Children’s Hospitals Network, Westmead, Sydney 2145, Australia; 3Specialty of Genomic Medicine, Faculty of Medicine and Health, University of Sydney, Westmead, Sydney 2145, Australia; elizabeth.farnsworth@health.nsw.gov.au (E.F.); bruce.bennetts@health.nsw.gov.au (B.B.); 4Sydney Genome Diagnostics, Western Sydney Genetics Program, Sydney Children’s Hospitals Network, Westmead, Sydney 2145, Australia; 5Specialty of Ophthalmology, Faculty of Medicine and Health, University of Sydney, Sydney 2006, Australia; 6Rare Diseases Functional Genomics Laboratory, Sydney Children’s Hospitals Network and Children’s Medical Research Institute, Westmead, Sydney 2145, Australia; dparekh@cmri.org.au (D.P.); lisa.riley@health.nsw.gov.au (L.G.R.); 7Specialty of Child and Adolescent Health, University of Sydney, Westmead, Sydney 2145, Australia; 8Stem Cell Medicine Group, Children’s Medical Research Institute, University of Sydney, Westmead, Sydney 2145, Australia; agonzalez-cordero@cmri.org.au; 9School of Medical Sciences, Faculty of Medicine and Health, University of Sydney, Sydney 2006, Australia

**Keywords:** *RPGR*, inherited retinal disease, rod-cone dystrophy, ciliopathy, iPSC, retinal organoids, retinal pigment epithelium

## Abstract

The *RPGR* gene encodes Retinitis Pigmentosa GTPase Regulator, a known interactor with ciliary proteins, which is involved in maintaining healthy photoreceptor cells. Variants in *RPGR* are the main contributor to X-linked rod-cone dystrophy (RCD), and *RPGR* gene therapy approaches are in clinical trials. Hence, elucidation of the pathogenicity of novel *RPGR* variants is important for a patient therapy opportunity. Here, we describe a novel intronic *RPGR* variant, c.1415 − 9A>G, in a patient with RCD, which was classified as a variant of uncertain significance according to current clinical diagnostic criteria. The variant lay several base pairs intronic to the canonical splice acceptor site, raising suspicion of an *RPGR* RNA splicing abnormality and consequent protein dysfunction. To investigate disease causation in an appropriate disease model, induced pluripotent stem cells were generated from patient fibroblasts and differentiated to retinal pigment epithelium (iPSC-RPE) and retinal organoids (iPSC-RO). Abnormal RNA splicing of *RPGR* was demonstrated in patient fibroblasts, iPSC-RPE and iPSC-ROs, leading to a predicted frameshift and premature stop codon. Decreased RPGR expression was demonstrated in these cell types, with a striking loss of RPGR localization at the ciliary transitional zone, critically in the photoreceptor cilium of the patient iPSC-ROs. Mislocalisation of rhodopsin staining was present in the patient’s iPSC-RO rod photoreceptor cells, along with an abnormality of L/M opsin staining affecting cone photoreceptor cells and increased photoreceptor apoptosis. Additionally, patient iPSC-ROs displayed an increase in F-actin expression that was consistent with an abnormal actin regulation phenotype. Collectively, these studies indicate that the splicing abnormality caused by the c.1415 − 9A>G variant has an impact on RPGR function. This work has enabled the reclassification of this variant to pathogenic, allowing the consideration of patients with this variant having access to gene therapy clinical trials. In addition, we have identified biomarkers of disease suitable for the interrogation of other *RPGR* variants of uncertain significance.

## 1. Introduction

Inherited retinal diseases (IRD) are a genetically heterogeneous group of blinding conditions that mostly affect the photoreceptor cells. Recent population genomic data indicates that IRDs may affect at least 1:1500 people [1], with a variable range in the age of onset and severity of visual loss. Rod-cone dystrophy (RCD), which affects rod photoreceptor cells with associated cone abnormality, is the most frequently occurring IRD. For patients with RCD, symptoms manifest in decreased night and peripheral vision, decline in visual acuity and progressive vision loss, leaving individuals with enormous physical, social and financial burdens. While the genetic causes of RCD are heterogeneous, with more than 60 identified contributing disease genes [2], pathogenic variants in Retinitis Pigmentosa GTPase Regulator (*RPGR*) are the main cause of X-linked RCD, and they cause approximately 20% of all RCD cases [3].

In retinal cells, *RPGR* undergoes alternate RNA splicing to express two main protein isoforms: the ubiquitously expressed RPGR^Ex1–19^ and RPGR^ORF15^, which is unique to the retina. RNA splicing is a highly conserved process wherein introns are removed by spliceosomes, which typically recognise introns by a 5′ donor “GT” site and a 3′ acceptor “AG” site. In *RPGR*, both isoforms are identically spliced up to exon 15. At this point, RPGR^Ex1–19^ transcribes exon 15 to exon 19, encoding a protein with 815 amino acids, while RPGR^ORF15^ splices in 1554 base pairs of intron 15, in addition to the canonical exon 15, and terminates, forming a protein with 1152 amino acids. *RPGR*-associated disease can be caused by missense, in-frame insertion/deletion, frameshift and splice-site variants located throughout exons 1–14 and can lead to nonsense-mediated decay and loss of function of the RPGR protein. In addition, insertion and deletion variants in the mutational hotspot region, ORF15, may result in protein truncation and more severe clinical presentations [4,5]. Approximately 70% of pathogenic *RPGR* variants result in predominately rod photoreceptor disease, while cone or cone–rod dystrophies are found in the remainder [6,7]. Variants in exon 1-14 and the proximal part of ORF15 are more frequently associated with rod-dominated disease, while those more in the 3′ region of ORF15 are more commonly associated with a cone or cone–rod phenotype [6]. Splice variants generally reflect this pattern, with studies indicating that correct splicing of both isoforms is integral for normal RPGR function [8,9,10].

Both RPGR isoforms localise to the primary cilium of the photoreceptor cells, where they bind and complex with transitional zone (TZ) proteins that are involved in a highly conserved intracellular trafficking system, known as intraflagellar transport (IFT) [11,12]. This mechanism allows the transport of cargo along the cilium and is essential for maintenance of healthy photoreceptor cells. The primary cilium of the inner segment (IS) is specially augmented to form a sensory structure known as the outer segment (OS), which is packed with photosensitive opsin discs that detect light. Due to high levels of ultraviolet radiation exposure and oxygen toxicity in the retina, opsin disc turnover rates are high, and the OS depends on the IFT system to deliver proteins manufactured in the IS for continued disc replenishment and light detection. RPGR forms complexes with other TZ proteins in the photoreceptor cilia, such as CEP290, to regulate the entry of soluble proteins and aid in the necessary cargo trafficking to maintain the OS [13,14]. Mutations throughout *RPGR* are known to impact the integrity of these interactions, leading to ciliopathy, cell death and ultimately, blindness [3,8,9,15,16,17].

In this study we report a novel intronic *RPGR* c.1415 − 9A>G variant in a male patient diagnosed with RCD. This variant was initially reported as a variant of uncertain significance (VUS). To investigate its pathogenicity, assays to assess splicing were performed on the patient’s cells. Further characterisations of *RPGR* expression and protein functional studies were performed using patient fibroblasts, as well as patient-derived induced pluripotent stem cells (iPSCs) differentiated into retinal pigment epithelial cells (iPSC-RPE) and retinal organoids (iPSC-ROs). We found decreased *RPGR* expression across these cell types, with decreased RPGR protein in the patient’s cells and RPGR loss at the TZ of primary and photoreceptor cilia. Additionally in the patient’s iPSC-ROs, there was rhodopsin and L/M opsin mislocalisation, increased photoreceptor apoptosis and dysfunction in actin regulation. Taken together, these findings allowed re-classification of the intronic variant in *RPGR* to a pathogenic variant and identified approaches in biomarker identification of variant pathogenicity, critical for patient eligibility for *RPGR* gene therapy clinical trials and other management purposes.

## 2. Materials and Methods

### 2.1. Patient Sample Collection and Participation

This study was approved by the Human Research Ethics Committee (HREC) of the Sydney Children’s Hospitals Network, Sydney, Australia. Written informed consent was obtained. Blood collection and skin biopsy from the male proband was undertaken.

### 2.2. Patient Ophthalmic Investigations

Ophthalmic investigations were undertaken in a 34-year-old Caucasian male (proband) and included visual acuity assessment, ultra-widefield pseudocolour fundus photos and autofluorescence (Optos plc, Dunfermline, UK). A full-field electroretinogram and pattern electroretinogram were performed to the standards of the International Society of Clinical Electrophysiology of Vision (ISCEV). The macular was assessed with Spectral Domain Optical Coherence Tomography (Heidelberg Engineering, Heidelberg, Germany).

### 2.3. Exome Sequencing, Analysis and In Silico Splice Prediction

The proband (Patient III-2, Figure 1A) underwent a TruSight One Clinical Exome (Illumina, San Diego, CA, USA) analysis through the Molecular Genetics Department, Sydney Genome Diagnostics, at the Sydney Children’s Hospitals Network (Westmead, Australia). As in our previous studies [18,19], the TruSight One Clinical Exome analysis has an average depth of coverage at 160x, and it is filtered to exclude regions with a <15x read depth. The allele frequency cut-off was >0.01. Subsequent reads were aligned to the hg19 reference genome and further filtered by examination against a gene panel of 68 known RCD disease genes. This analysis identified a novel variant in RPGR: c.1415 − 9A>G. An in-silico pathogenicity analysis was undertaken on the novel variant using Alamut Visual Software (Version 2.5, Interactive Biosoftware, Rouen, France), as per our previous studies [18]. The variant was assessed using the Alamut Visual splicing prediction tool, which amalgamated algorithmic data from five distinct splice prediction programs (SSF, MaxEnt, NNSPLICE, GeneSplicer and HSF) where consensus across the algorithms was required for the variant to be considered significant. Variant confirmation and segregation studies were performed on PCR amplicons by bi-directional Sanger sequencing on an ABI3730xl instrument, undertaken at the Australian Genome Research Facility (Westmead, Australia). Variant classification was in accordance with established American College of Medical Genetics and Genomics (ACMG, Bethesda, MD, USA) guidelines [20]. 

### 2.4. RPGR Gene Expression Studies

Total RNA was extracted from cells using the RNeasy micro kit (Qiagen, Germantown, MD, USA), according to the manufacturer’s instructions. Total RNA was converted to cDNA with the SuperScript IV First-Strand Synthesis System (Invitrogen, Waltham, MA, USA). Primer pairs were designed using Primer-BLAST (NIH, Bethesda, MD, USA) against reference sequences RPGR^Ex1–19^, NM_000328.2, and RPGR^ORF15^, NM_001034853.1, which are identical in the regions under investigation in this study. For splicing assays, the Sanger sequencing of control and patient cDNA was used to interrogate *RPGR* splicing across the region of the variant under investigation, using two separate sets of PCR primer pairs. The first set of primer pairs was in exons 11 and 14: FWD 5′AGGGACTCTTGGCCTTTCTG3′ and RVS 5′GTATCCTGCGTCAGTTCCCC3′. A second set of primer pairs was placed in exons 10 and 14: FWD 5′GACTCTATCAGCACGTATGCG3′ and RVS 5′GCTGCGTCATGAAAATCCCTTG3′. Standard PCR conditions were used for amplification, and PCR products were Sanger sequenced (Macrogen, Seoul, South Korea). To quantitate relative gene expression levels of *RPGR,* an RT-qPCR was undertaken using primer pairs in exons 1 and 3, FWD 5′CCGACCAAACCGTCCTCTAC3′ and RVS 5′TGACCCCAGTTGTTACTGCC3′, and exons 11 and 12, FWD 5′GACCTCATGCAGCCAGAG3′ and RVS 5′GTTTCTCCAAGGCTTTCTACAG3′. *POLR2A* and *HPRT* primers were used as housekeeper control genes in all cell types [21].

### 2.5. Cell Culture and Generation of iPSCs

A primary fibroblast cell line was created from the proband (Patient III-2, Figure 1A) using previously established protocols [22]. Fibroblasts were maintained in Minimum Essential Medium Eagle (MEM) containing 15% foetal calf serum and 1x Non-Essential Amino Acids (NEAA), and they were incubated at 37 °C and 5% CO_2_. Cells were passaged at 80% confluency using TrypLE Select (1X) (Gibco ThermoFisher Scientific, Waltham, MA, USA).

The generation of iPSCs from patient III-2 fibroblasts was performed by the stem cell facility, StemCore (Australian Institute for Bioengineering and Nanotechnology, University of Queensland, Brisbane, Australia), and as previously described for normal skin fibroblasts (European Collection of Authenticated Cell Cultures (ECACC), Salisbury, UK) [21], (Control 1). A second control iPSC line, HPSI0314i-hoik_1 (Control 2), was purchased from ECACC. iPSCs were maintained on Matrigel (Corning, Corning NY, USA)-coated 6-well plates in mTeSR1 (StemCell Technologies, Vancouver, Canada) media.

### 2.6. iPSC Pluripotency and Trilineage Differentiation

Pluripotency and trilineage differentiation capability of the patient iPSC lines was carried out as previously described for the Control 1 line [21]. Pluripotency characterisation was undertaken through RT-qPCR, using primers amplifying pluripotency markers *OCT4*, *NANOG* and *SOX2* (Appendix A), as well as an immunofluorescence assessment [21]. For karyotyping, cells were maintained until being 60% confluent before incubation with KaryoMAX colcemid (10 µg/mL) (ThermoFisher, Sydney, Australia). Cells were then washed and dissociated to single cells and incubated with 0.06 M KCl. Cells were fixed in fresh Carnoy’s Fixative (3-parts methanol to 1-part glacial acetic acid) 3 times before being suspended, dropped onto glass slides and stained with Leishman’s stain. Subsequent metaphases were imaged using the iKaros software (MetaSystems, Altlussheim, Germany).

For assessment of trilineage differentiation, briefly, embryoid bodies (iPSC-EBs) were generated from detached cell aggregates of confluent iPSC cultures placed in a suspension in E8 media. After 24 h, the iPSC-EBs were switched to spontaneous differentiation media (KnockOut Dulbecco’s Modified Eagle’s Medium, 20% KnockOut serum replacement, 1 × NEAA, 1 mM Glutamax, 100 µM B-mercaptoethanol, and 1 × Penicillin Streptomycin) prior to plating on Matrigel-coated cells. At day 14, the iPSC-EBs were collected for RT-qPCR to assess the induced expression of markers of the three germ layers using primers previously described [21].

### 2.7. Differentiation of iPSCs to Retinal Cells

The differentiation of iPSCs to retinal cells was undertaken at 95% confluency using a proneural induction method previously described [23]. iPSC-RPE islands and retinal vesicles were manually excised from day 40–60 differentiating cultures and maintained on Matrigel-coated culture wells in retinal differentiation media (1 × B27 without Vitamin A, 1 × Penicillin Streptomycin, DMEM/F12). Vesicles were placed in suspension in 96-well ultra-low attachment plates in retinal differentiation media [23] and allowed to mature to iPSC-ROs.

### 2.8. Preparation of Samples for Immunofluorescence, Cilia and TUNEL Assays and Immunohistochemistry

All cells for cilia and immunofluorescence assays were plated with an equal cell number onto prepared coverslips in 24-well plates. Fibroblast cells additionally were starved in 1x NEAA in MEM for 24 h prior to fixation with 4% PFA or methanol. Cells were blocked in 0.1% fish gelatine and 0.02% Triton X-100 in PBS before overnight staining with primary antibodies: γ-tubulin (T5326, Sigma-Aldrich, St. Louis, MO, USA, 1:200), anti-RPGR (HPA001593, Sigma-Aldrich, St. Louis, MO, USA, 1:250), acetylated α-tubulin (T6793, Sigma-Aldrich, St. Louis, MO, USA, 1:1000) and ZO-1 (40-2200, Life Technologies, Carlsbad, CA, USA, 1:200). Secondary antibodies were applied for 2 h in the dark (anti-mouse Alexa Fluor-594, 1:1000; anti-rabbit Alexa Fluor-488, 1:1000; DAPI, ThermoFisher Scientific, Waltham, MA, USA). To obtain the cilia length in fibroblasts and iPSC-RPE cells, 100 cell positions were randomly chosen for imaging on an LSM 880 Airyscan Confocal microscope (Zeiss, Jena, Germany). Cilia length was measured using the Zen Blue software (Zeiss, Jena, Germany), and the rate of ciliated cells was calculated as a percentage of total number of cells imaged.

iPSC-ROs were embedded, either unfixed or fixed, in 4% PFA followed by 20% sucrose/PBS, in an Optimal Cutting Temperature Compound (O.C.T, Sakura Tissue-Tek^®^), then cryosectioned for immunohistochemical staining. For apoptotic investigations, sectioned iPSC-ROs were prepared according to the manufacturer’s guideline for the DeadEnd^TM^ Fluorometric TUNEL System (G3250, Promega, Madison, WI, USA). For immunohistochemistry investigations, sections were blocked and stained as described for the cellular assays using the following antibodies against: RPGR (HPA001593, Sigma-Aldrich, St. Louis, MO, USA, 1:400), CEP290 (SANTSC-390462, Santa Cruz Biotechnology, Dallas, TX, USA, 1:200), Phalloidin (A30107, ThermoFisher Scientific, Waltham, MA, USA, 1:500), Rb x Opsin Red/Green (L/M opsin) (AB5405, EMD Millipore, Burlington, MA, USA, 1:100), Anti-opsin (Rhodopsin) (O4886, Sigma-Aldrich, St. Louis, MO, USA), Rootletin (SANTSC-374056, Santa Cruz Biotechnology, Dallas, TX, USA, 1:500), Pericentrin (ab28144, Abcam, Cambridge, UK, 1:500), CRX (H00001406-M02, Sapphire Bioscience, Sydney, Australia, 1:300) and recoverin (AB5585; EMD Millipore, Burlington, MA, USA, 1:1000).

### 2.9. Western Blot Analysis

For the western blot, protein (30 µg) was loaded into a NuPAGE 3–8% Tris-Acetate gel (Invitrogen, Waltham, MA, USA). Gels were transferred to a 0.45 µm nitrocellulose membrane for 1.5 h at 30 V in the NuPAGE transfer buffer at room temperature. Membranes were probed overnight at 4 °C with 1:500 anti-RPGR antibody (HPA001593; Sigma-Aldrich, St. Louis, MO, USA) with the following immunogen sequence, encoded by exons 10 to 12: 379EINDTCLSVATFLPYSSLTSGNVLQRTLSARMRRRERERSPDSFSMRRTLPPIEGTLGLSACFLPNSVFPRCSERNLQESVLSEQDLMQPEEPDYLLDEMTKEAEIDNSSTVESLGETTDILNMTHIMSLN509, and they were used in previous studies for western blot [24,25]. This region is common across the RPGR^Ex1–19^ and RPGR^ORF15^ protein isoforms. Normal human retinal protein was used as a positive control. Studies with the Sigma antibody were optimised alongside an RPGR antibody (16891-1-AP; Proteintech, Rosemont, IL, USA) provided for western blot analysis, as well as lysates from normal human retina and control human iPSCs. Vinculin (ab130007; Abcam, Cambridge, UK) was used as a loading control. The membrane was imaged using the Odyssey Fc system (LI-COR Biosciences, Lincoln, NE, USA).

### 2.10. Image Analysis, Total Fluorescenc, and Statistics

The fluorescence intensity analysis, co-localization analysis and integrated density analysis were carried out using the FIJI software, with readings for the area, area-integrated density, mean grey value and background collected [26]. For total fluorescence (TF), the regions of interest (ROI) were defined using the Zen Blue software (Zeiss, Jena, Germany) against the DAPI channel, and they were exported against the channel of interest for each image analysed. The regions chosen in iPSC-ROs had a clear distinction between the outer nuclear layer (ONL) and inner nuclear layer (INL), with little to no core debris. TF was calculated as Integrated density—(area of ROI x average background fluorescence). For apoptosis analysis, the ONL was defined against the DAPI channel using the Zen Blue software, and TUNEL+ cells were counted manually in these ROI and normalised against area. For L/M opsin+ cell counts, five ROIs of the same size were chosen in areas of the image with distinction between the ONL and INL. Photoreceptor cells were counted as L/M opsin+ if there was staining of the IS + O, or if there was staining of the soma. The analysis of data was done with the GraphPad Prism 5 software. Paired and unpaired Student’s t-tests were used where appropriate. The standard error of mean (SEM) was calculated for error values. A minimum of three independent experiments were carried out for statistical analyses.

## 3. Results

### 3.1. Patient Ophthalmic Features of RCD

The proband (III-2, Figure 1A) was noted to have impaired night vision from 6 years of age and was diagnosed with RCD at 11 years of age. On review at 34 years of age, he had Visual acuity on the right of 6/12 + 1 and left of 6/9 − 1, with a reading acuity of N5 in each eye. The pattern electroretinogram (ERG) and full-field ERG signals were undetectable, indicating both rod and cone photoreceptor cell dysfunction. Ultra-widefield pseudocolour fundus photos showed retinal atrophy across the fundus, with mild pigmentary disturbance in the mid periphery (Figure 1B). Ultra-widefield fundus autofluorescence imaging identified a narrow ring of hyperautofluorescence around the fovea. Beyond the vascular arcades, there were patchy areas of hypoautofluorescence scattered throughout the fundus (Figure 1C). Macular spectral domain optical coherence tomography scans showed only a residual ellipsoid zone at the fovea in each eye (Figure 1D).

### 3.2. Exome Sequencing, Segregation and Bioinformatic Analysis Identified a Novel Intronic RPGR Variant

To investigate the genetic cause of the RCD phenotype, the proband genomic DNA was collected and analysed through TruSight One Clinical Exome Sequencing. A hemizygous variant (NM_000328.2 and NM_001034853.1; c.1415 − 9A>G) in intron 11 of *RPGR* was identified (Figure 1E). This variant was not previously reported and is absent from gnomAD (version 2.1.1). The diagnostic report by the genomic laboratory classified this as a VUS. The proband’s mother (II-1, Figure 1A) and maternal aunt (II-2, Figure 1A) were found to be heterozygous carriers of the same novel variant, while his unaffected brother (III-1, Figure 1A) did not have the variant. The family pedigree indicated an X-linked inheritance pattern. To establish a possible splicing abnormality, the Alamut Visual software v2.13 (Interactive Biosoftware, Rouen, France) was employed, and the analysis returned a high probability that the novel A>G variant would create an upstream alternate AG splice acceptor site and abolish the function of the canonical acceptor site (Figure 1F). This would be predicted to result in abnormal splicing and the inclusion of 8 bp of an intronic sequence between exons 11 and 12 of the *RPGR* transcripts, including the two main isoforms encoding RPGR^Ex1–19^ and RPGR^ORF15^.

### 3.3. Novel RPGR Intronic Variant Creates an Alternate Splice Acceptor Site and Loss of RPGR Expression, Especially at the Transition Zone in Fibroblast Primary Cilia

To examine this predicted splicing defect, the RT-PCR analysis of the patient fibroblast cDNA in the variant region of interest (primer pairs in exons 11 and 14) followed by Sanger sequencing identified the abnormal presence of an additional eight base pairs between the exon 11 and exon 12 junction (Figure 2A). The product amplified using primer pairs in exons 10 and 14 verified these results, corroborating the *in-silico* Alamut prediction and confirming abnormal splicing. The addition of eight base pairs causes a frameshift, predicted to alter the subsequent six amino acids before terminating in a premature stop codon early in exon 12, p.(Asp472Valfs*7). Such an alteration in the sequence is predicted to result in a truncated protein, with a potential for nonsense-mediated decay of *RPGR* mRNA transcripts.

The impact of the novel *RPGR* variant on gene expression was initially explored in patient fibroblast cells. Two regions of *RPGR* were examined using RT-qPCR, with primers binding to exons 1–3 and exons 11–12. Both primer sets demonstrated a significant decrease in *RPGR* expression in the patient fibroblasts when compared to control cells (exons 1–3: * *p* < 0.05; exons 11–12: * *p* < 0.05; *n* = 3 independent experiments) (Figure 2B).

To better understand the implications of the RT-qPCR findings, RPGR protein expression was visualised through the immunofluorescence analysis of fibroblasts. Typically, RPGR localises largely to the TZ of the cilium, where it interacts and complexes with other ciliary proteins, with localisation also seen at the basal bodies and the axoneme of the cilium [12,27,28]. In control fibroblasts, we found RPGR protein localisation most obviously at the TZ of fibroblast cilia (Figure 2C(i), white arrow), whereas this was decreased in the patient fibroblast cilia (Figure 2C(ii), white arrow outline). Quantification by counting the number of cilia with RPGR staining in the TZ as a percentage of the total number of cilia present confirmed a marked reduction of RPGR expression at the TZ in the patient compared with control fibroblasts (** *p* < 0.01; *n* = 159 control cilia counted; *n* = 219 patient cilia counted) (Figure 2C(iii)). However, quantification of ciliogenesis in these cells showed no changes in the percentage of ciliated cells, and measurement of the cilia in these cells showed a similar cilia length between the patient and control fibroblasts (Appendix A).

### 3.4. Control and Patient iPSC Lines Differentiate into iPSC-RPE and iPSC-ROs

To delineate the effects of the novel *RPGR* variant in the relevant retinal tissue, the patient’s (Patient III-2) fibroblasts were reprogrammed into iPSCs for differentiation to iPSC-RPE and -ROs alongside two control iPSC lines. Two patient iPSC clonal lines (Patient Clone 1 and Patient Clone 2) that were generated were shown by RT-qPCR to express pluripotency markers *OCT4*, *NANOG* and *SOX2* compared to the fibroblasts (Figure 3A). The iPSCs were further characterised for expression of pluripotent-specific proteins NANOG and SSEA-4, and they were shown to have normal karyotypes (Figure 3B). The trilineage differentiation of iPSC-EBs formed from the patient iPSCs showed an increased expression of *AFP*, *CDH20*, *EN1*, *FOXF1*, *HAND2* and *PHOX2B* compared to iPSCs, indicating the cells’ ability to differentiate into the endoderm, mesoderm and ectoderm germ layers (Figure 3C).

Prior to retinal cell differentiation, genomic DNA from patient iPSC lines was examined by Sanger sequencing and shown to confirm the presence of the *RPGR* c.1415 − 9A>G variant (data not shown). For iPSC-RPE and iPSC-ROs, the two patient clonal iPSC lines and two control iPSC lines (Control 1 and Control 2) were differentiated using previously described retinal induction methods [23]. By 4–6 weeks of development, bright-field images confirmed that all the control and patient iPSC lines were proficient in forming pigmented iPSC-RPE islands (Figure 3D(i)). The immunohistochemistry analysis showed ZO-1 positive iPSC-RPE cells with a typical hexagonal cell shape and the presence of cone–rod homeobox transcription factor CRX (Figure 3D(ii)). Similarly, all lines were also proficient in forming laminated iPSC-Ros (Figure 3E(i)) with CRX and recoverin positive photoreceptor cells in ONL-like structures (Figure 3E(ii)), which mature and ciliate to form the photoreceptor OS.

### 3.5. RPGR Variant iPSC-RPE Exhibits a Loss of RPGR Localisation at the Transitional Zone of Primary Cilia

*RPGR* mRNA from patient iPSC-RPE cells was examined by RT-PCR Sanger sequencing and showed the splicing abnormality caused by the *RPGR* c.1415 − 9A>G variant, as seen in patient fibroblast cells (Appendix A). To determine the expression of *RPGR* in iPSC-RPE cells, we performed a RT-qPCR analysis. Similar to the data observed in patient fibroblasts, *RPGR* expression was significantly decreased in the patient’s iPSC-RPE when compared to control iPSC-RPE cells (** *p* < 0.01, *** *p* < 0.001; *n* = 4 independent experiments) (Figure 4A). These results were observed using both exon 1-3 and exon 11-12 primer sets. Similar to the fibroblast cells, ciliogenesis was calculated as a percentage of cells with acetylated α-tubulin-positive cilia, and the cilia length was measured. Control and patient iPSC-RPEs showed no difference in ciliogenesis or cilia length (Appendix A).

Consistent with reduced *RPGR* transcripts in patient iPSC-RPE cells, the immunofluorescence analysis identified lower levels of RPGR protein co-localisation with acetylated α-tubulin-positive TZ in the cilia of both patient clones in the iPSC-RPE. In both control iPSC-RPE cell lines, there was a presence of RPGR staining at the TZ of the primary cilium (Figure 4B(i),C(i), white arrow, yellow region in merged images). In contrast, in both patient clone iPSC-RPE cell lines, there was a lack of RPGR staining in the TZ region of the cilium (Figure 4B(ii),C(ii), white arrow outline). Quantification of the number of cilia expressing RPGR staining at the ciliary TZ confirmed the decrease of RPGR in the patient TZ region of the cilia compared with the control (**** *p* < 0.0001, *n* = 159 control cilia counted, *n* = 219 patient cilia counted) (Figure 4B(iii)). RPGR staining was also noted in punctate regions near the ciliary TZ in the control and patient iPSC-RPE cells (Figure 4B(i),(ii), paired white arrowheads). To determine the localisation of the RPGR punctate staining, iPSC-RPEs were also co-stained with the basal body marker pericentrin (PCN) (Figure 4C(i),(ii), red punctate staining, white asterisks), and this co-staining with PCN indicated that the punctate RPGR staining was in the basal bodies of the primary cilia of both control and patient iPSC-RPEs (Figure 4C(i),(ii), paired white arrowheads overlapping asterisk).

### 3.6. RPGR Variant iPSC-ROs Have Decreased RPGR Expression at the Ciliary TZ of the Photoreceptor Cells

We next investigated RPGR expression and presence in iPSC-RO photoreceptor cells. Control iPSC-ROs at 30 weeks formed photoreceptor cells with OS and displayed typical RPGR localisation apically to the ciliary rootlet (Figure 5A). This indicated that the RPGR protein, similarly to fibroblasts and iPSC-RPE, localises to the ciliary TZ in photoreceptor cells. In mRNA from mature iPSC-ROs, we showed the splicing abnormality caused by the RPGR c.1415 − 9A>G variant, as seen in patient fibroblasts and iPSC-RPE (Appendix A). Similar to previous results, the RT-qPCR analysis of iPSC-ROs using exon 1-3 and exon 11-12 primer sets showed a significant decrease in *RPGR* gene expression in patient iPSC-ROs when compared to control iPSC-ROs (** *p* < 0.01, *n* = 4 independent experiments) (Figure 5B).

To evaluate if the reduced transcript levels affected protein expression, we used western blotting to examine the RPGR protein in control and patient iPSC-ROs. Our results demonstrated a clear decrease in the RPGR protein in patient-derived iPSC-ROs compared with the control (Figure 5C). The antibody immunogen sequence is expected to bind to both major retinal isoforms of *RPGR*. RPGR^Ex1–19^ has a predicted weight of 90 kDa. Previous work in the retina has proposed that bands seen at 110, 140, 150 and 160 kDa are caused by post-translational modifications to the protein [29], while a band at 125 kDa has been shown to be modulated in the presence of translational read-through therapy for RPGR nonsense mutations [25]. RPGR^ORF15^ is expected to have a molecular weight of 127 kDa, with alternate splicing and post-transcriptional protein modifications influencing migration rates, causing additional bands at 100, 120, 140 and 200–250 kDa [29,30], with a 220 kDa band present in HEK293T cells transfected with wildtype and codon-optimised RPGR^ORF15^ [24]. In this study, control iPSC-ROs displayed a multi-banded western blot appearance typical of RPGR, but in the patient iPSC-ROs, these multiple bands were absent, consistent with the reduction of expression of both RPGR^Ex1–19^ and RPGR^ORF15^ as expected. However, there was still some protein expression evident, albeit at reduced levels, at the 110 kDa band and 290 kDa band.

The ciliary protein CEP290 localises to the TZ in photoreceptor cilia and is known to interact with RPGR [31,32]. In both control 30-week-old iPSC-ROs, there was strong co-localisation of CEP290 and RPGR at the photoreceptor TZ (Figure 5D(i) and Appendix A, white arrows, yellow co-stain in merged images). In patient iPSC-ROs from both clones, there was a decreased RPGR expression at the TZ of the photoreceptor cilium (Figure 5D(ii) and Appendix A, white arrow outline), consistent with our western blot experiments and aligning with data observed in patient fibroblasts and iPSC-RPEs. The co-localization between RPGR and CEP290 staining along the patient iPSC-RO photoreceptor ciliary TZ was greatly reduced (**** *p* < 0.0001; *n* = 4 independent iPSC-ROs) (Figure 5D(iii)). There was some RPGR protein expression seen in the iPSC-RO away from the CEP290-stained photoreceptor TZ in both control and patient iPSC-ROs (Figure 5D(i),(ii) and Appendix A, paired arrowheads).

### 3.7. RPGR Variant iPSC-ROs Display Mislocalised Opsins, Increased Photoreceptor Apoptosis and Abnormal F-Actin Expression

*RPGR*-associated disease affects both rod and cone photoreceptor cells, with mislocalisation of both rod and cone opsins seen in murine and canine models of RPGR disease [33,34]. Patient III-2 displayed clinical features of disease affecting both these cell types; hence, we next investigated rhodopsin and L/M opsin staining in control and patient iPSC-ROs. In all iPSC-ROs examined, rhodopsin staining was present at the IS + OS region of the rod photoreceptor cells as expected (Figure 6A(i),(ii) and Appendix A, white arrowheads). However, in the patient iPSC-ROs, rhodopsin staining appeared to be somewhat increased in the soma of the photoreceptor cells (ONL) (Figure 6A(ii),B(ii) and Appendix A, white arrows), compared to control iPSC-ROs (Figure 6A(i),B(i), white arrow outline). To quantify this, for each iPSC-RO, the TF of rhodopsin staining was examined (Appendix A) in the ONL region of the rod photoreceptor cells and compared with the TF of rhodopsin staining in the IS + OS region by expressing the values as a ratio. Using this approach, rhodopsin localisation in the ONL of the patient iPSC-ROs was increased compared to control iPSC-ROs (Figure 6A(iii)) (* *p* < 0.05, *n* = 3 independent iPSC-ROs).

In typical iPSC-ROs, L/M opsin is observed in the IS + OS region of cone photoreceptor cells, and in the soma and axons of cone photoreceptors (ONL) of the maturing control iPSC-ROs (Figure 6C(i) and Appendix A, white and red arrowheads) [23,35,36]. This pattern of protein localisation was also observed in patient iPSC-ROs (Figure 6C(ii) and Appendix A, white and red arrowheads). However, quantification of the number of L/M opsin+ cells showed a decrease in the number of cone photoreceptor cells in the outer region (IS + OS + ONL) of patient iPSC-ROs compared to controls (** *p* < 0.01, *n* = 3–5 independent iPSC-ROs) (Figure 6C(iii)). In control and patient iPSC-ROs, there was additional L/M opsin localisation in the inner regions of the iPSC-RO, namely in the outer plexiform layer (OPL) and the INL region (Figure 6C(i),(ii) and Appendix A, white arrows). Despite a decrease in L/M opsin+ cone photoreceptor cell presence in the outer regions of the patient iPSC-ROs, we noted that the overall total number of L/M opsin+ cells, inclusive of cells in the inner regions (OPL and INL) of the iPSC-ROs, appeared similar between patient and control iPSC-ROs. This implied that there may be an increase in the number of cone photoreceptor cells mislocalised to the inner regions of the patient iPSC-ROs, and expression of the number of L/M opsin+ cell soma staining in the outer versus the inner regions (IS + OS + ONL vs. OPL + INL) of the iPSC-ROs confirmed this (** *p* < 0.01, *n* = 3–5 independent iPSC-ROs) (Figure 6C(iv)). This finding was reflected in the accumulation of L/M opsin staining seen in the inner region of patient iPSC-ROs (Figure 6C(ii) and Appendix A, white arrows). Additionally, we noticed non-nuclear L/M opsin staining in the OPL region of patient iPSC-ROs, where photoreceptor pedicles are found (Figure 6C(ii) and Appendix A, white arrow outlines). To quantify this, we looked at the TF values of the OPL and INL, expressed as a ratio with the outer region of the iPSC-RO, in order to analyse and compare fluorescence between images (Appendix A). Patient iPSC-ROs showed a higher proportion of L/M opsin TF in the inner region versus the outer region of the iPSC-RO when compared to the controls (* *p* < 0.05, *n* = 3–5 independent iPSC-ROs) (Figure 6C(v)), indicating that more L/M opsin protein was present in the inner regions of patient iPSC-ROs.

Previous investigations have indicated increased apoptosis in the ONL of canine and iPSC-RO models of retinal dystrophies with rhodopsin defects and mislocalisation [36,37]. In our study, the TUNEL assay in the patient iPSC-ROs indicated higher levels of apoptotic nuclei present in the ONL compared to control iPSC-ROs (Appendix A) (* *p* < 0.05, *n* = 3 independent iPSC-ROs).

Previous studies in an RPGR mouse model and patient-derived iPSC-ROs have demonstrated the upregulation of F-actin associated with pathogenic RPGR mutations [38]. RPGR has been implicated in actin regulation in cells through protein interactions that affect actin cleavage and polymerisation [30,39]. Therefore, to further confirm the pathogenicity of the novel *RPGR* c.1415 − 9A>G variant, we investigated potential disruptions to RPGR interactions by examining F-actin levels in control and patient iPSC-ROs via phalloidin staining (Figure 7). Increased phalloidin staining was detected in patient iPSC-ROs, particularly at the outer limiting membrane (OLM) (Figure 7i,ii), and quantification of the pixel intensity confirmed a two-fold increase in phalloidin intensity compared to control iPSC-ROs (* *p* < 0.05, *n* = 4 independent iPSC-ROs) (Figure 7iii). Overall, these results demonstrate that *RPGR* c.1415 − 9A>G iPSC-ROs have an abnormal phenotype. Taken together with the data obtained from patient fibroblasts and iPSC-RPE, our study supports the reclassification of the novel *RPGR* c.1415 − 9A>G variant as pathogenic, and it highlights the utility of precision medicine approaches including the use of iPSCs and their derivatives.

## 4. Discussion

*RPGR* is a well-known retinal dystrophy disease-causing gene, but VUSs may be identified in this gene, which hamper clinical management decisions and access to clinical trials. In this study, we have demonstrated the value of patient-derived retinal tissues for genetic variant investigation. The affected patient in this study was diagnosed with RCD and had a family history consistent with X-linked recessive inheritance, and he was found to have an intronic VUS in *RPGR*, c.1415 − 9A>G. The bioinformatic analysis was indicative of a splicing abnormality, raising the requirement for variant investigation in tissues where RPGR is expressed and, ideally, in retinal tissues where the disease manifests. Hence, we created patient-derived iPSC lines for differentiation to iPSC-RPE and -ROs for investigation of this variant, in addition to investigation in patient fibroblasts. This study demonstrated that the patient variant c.1415 − 9A>G within intron 11 of *RPGR* forms an alternate splice acceptor site just upstream of the canonical splice acceptor site, resulting in the inclusion of eight base pairs at the 3′ end of intron 11 in all cell types examined. This results in a predicted frameshift and truncation of the protein in exon 12, affecting both the RPGR^Ex1–19^ and RPGR^ORF15^ isoforms, and it is also expected to lead to nonsense-mediated decay of both isoforms. We confirmed this by demonstrating a decreased *RPGR* expression at both the mRNA and protein levels in the patient fibroblasts, iPSC-RPE and iPSC-ROs. This work also demonstrated mislocalisation of rhodopsin and cone L/M opsin staining, photoreceptor apoptosis, and increased presence of F-actin as indicators of abnormality in the patient iPSC-ROs compared with controls. This study highlights the value of patient-derived retinal tissue for the establishment of assays and distinct morphological markers that can distinguish pathogenic *RPGR* variants from benign variants to enable patient eligibility for clinical trials and other clinical management avenues.

RPGR mutations in the shared sequence of RPGR isoforms are known to cause retinal disease phenotypes, including RCD, cone-rod dystrophy and cone dystrophy, with a predominance of the RCD phenotype in the exon 1-14 region [6,7]. The RCD phenotype in the patient in this study was consistent with the variant localisation in the exon 1-14 region of *RPGR*. The domains encoded by the shared sequence allow for protein binding and result in direct and indirect interactions with a number of cilia proteins implicated in the ciliary trafficking of membrane proteins, including rod and cone opsins [3,40]. Normal splicing of the protein in this region is therefore important for RPGR function in the retina [8,9,10,41], and it gives evidence to a pathogenic role of the novel *RPGR* c.1415 − 9A>G variant in patient cells through diminished protein functionality. In this study, abnormal splicing in patient cells, including fibroblasts, iPSC-RPE and iPSC-ROs, resulted in a decrease in *RPGR* expression, with immunofluorescence studies indicating a loss of RPGR presence most notably at the TZ of the primary cilium. Previous work in hTERT-RPE1 and IMCD3 cells established that a RPGR^Ex1–19^ prenylated C-terminal interaction with PDE6D is required for RPGR to localise to the TZ of the cilium [12]. Other studies suggest that prenylated RPGR^Ex1–19^ migrates at 140–160 kDa [29], and the western blot analysis showed the complete loss of this band in our patient iPSC-ROs. In addition to overall decreased expression of RPGR, loss of this particular isoform may further contribute to loss of expression at the ciliary TZ.

The loss of the RPGR protein at the TZ in the primary cilium of patient fibroblasts and iPSC-RPE, as well as the photoreceptor cells in patient iPSC-ROs, is a strong indicator of pathogenicity of the RPGR c.1415 − 9A>G variant. RPGR complexes with other IRD proteins at the cilium, including CEP290, which is required for ciliogenesis and normal cilia trafficking, and pathogenic variants in this gene also cause a range of syndromic retinal diseases. RPGR and CEP290 are known to form at least two distinct complexes at the photoreceptor ciliary TZ, with at least one of these complexes involved in the trafficking of other proteins, such as opsins, to the OS [31,42]. The decreased RPGR expression in our patient iPSC-ROs caused a loss of co-localization of RPGR with CEP290, indicating an inability of the patient RPGR protein to properly localise to the photoreceptor TZ and complex with other TZ proteins for normal function of the TZ region.

The demonstrated lack of expression of RPGR in the photoreceptor TZ complexes in patient iPSC-ROs suggests that there may be a potential interference with normal IFT function and trafficking of photoreceptor opsins. Rhodopsin is normally trafficked from the Golgi apparatus to the basal bodies, and from there, it is moved through the IFT to localise to the OS of rod photoreceptor cells [34,38]. While other RPGR iPSC-RO models of disease have shown changes in rod photoreceptor cell morphology and decreased numbers of rod cells [36,43], acquired images in our model system did not show distinct cells, and so this method of interrogation could not be pursued. However, we were able to show an increase in mislocalisation of the rhodopsin protein to the soma of the rod photoreceptor cells, which are in the ONL of the iPSC-ROs, through TF expression ratios. This finding suggests that more rhodopsin protein remains in the soma of the patient rod photoreceptor cells, implying a decrease or dysfunction in RPGR-associated IFT trafficking of rhodopsin to the OS of the photoreceptor cells, as noted in other models of RPGR-associated diseases [33,34,44].

Increased L/M opsin staining was also seen in the OPL in the patient iPSC-ROs, with some staining in structures suggestive of cone synapse pedicles, which may have been due to interference with the normal IFT function and mislocalisation of L/M opsin in the cone photoreceptor cells. This is consistent with the features identified in canine and murine RPGR^ORF15^ disease models, where L/M opsin and M opsin staining is increased at the axons and the synapse pedicles of cone photoreceptor cells [17,34], and it would suggest a degree of IFT dysfunction in our patient iPSC-RO photoreceptor cells. Co-staining with a cone pedicle marker would be helpful to confirm this phenotype. The majority of the increase in the L/M opsin staining in the OPL and INL in the patient iPSC-ROs appeared to be due to mislocalisation of cone photoreceptor nuclei and soma to this region. This phenotypic appearance may be seen in normal iPSC-ROs [23,35,36], and it may relate to developmental aspects of cone photoreceptor cell localisation in iPSC-ROs. Developmental impact of RPGR abnormality on cone photoreceptor cell localisation has not previously been recognised in other model systems, so further work is needed to determine the contribution of RPGR, or other factors, to this phenotypic appearance.

While RPGR expression was lost at the TZ in the primary cilium of patient fibroblasts and retinal cell types examined in this study, there remained some residual protein expression in the region of the basal bodies in the iPSC-RPE and likely in a similar region of the iPSC-RO photoreceptor cells. Previous studies have suggested the activity of cytoplasmic dynein-1 in retrograde microtubule transport for RPGR^ORF15^ localisation to basal bodies [28]. Cytoplasmic dynein-1 complexes with dynactin to bind to and transport protein vesicles from various parts of a cell, such as the nucleus or Golgi apparatus, to other cellular components, such as centrosomes and basal bodies [45]. RPGR is known to physically bind to two dynactin proteins that contribute to the dynein–dynactin complex [28]. The RPGR binding domains for these dynactins are encoded by RPGR exons 10–15. It is possible that residual truncated RPGR protein of both RPGR isoforms in the patient retinal and fibroblast models could maintain interaction with dynactin in the dynein–dynactin complex and be trafficked to the ciliary basal bodies from the Golgi apparatus, thus explaining why there may be residual RPGR staining at the basal bodies in patient retinal cells.

Photoreceptor apoptosis was noted in patient iPSC-ROs in this study. This is a significant phenotype in the XLPRA2 canine model, which is a severe RPGR-associated disease model with significant opsin mislocalisation and photoreceptor death [17]. Increased cell death as a phenotype in iPSC-ROs has also been noted in association with other variants in *RPGR* and variants in *RP2* [36,43], and so this may be a useful biomarker for the investigation of the pathogenicity of *RPGR* VUSs in future studies.

RPGR has also been identified as a positive modulator of the protein Gelsolin, which, when active, cleaves actin [38]. Additionally, in *RPGR* knockdown models, an increase in Dishevelled (DVL) proteins have been found to lead to increased active RhoA, a GTPase protein that polymerises actin, leading to an increase in actin filaments in both cell and animal models [30,39]. Patient iPSC-ROs show an increase in F-actin presence, and so the large losses of RPGR protein may impair the functional role of RPGR in gelsolin activation and DVL protein regulation. In this study, in patient-derived iPSC-ROs, there was an accumulation of F-actin, indicating that the novel c.1415 − 9A>G variant may also lead to cytoskeletal dysregulation. This provides another potential readout assay for pathogenicity testing of other *RPGR* VUSs. By having clear markers that indicate pathogenicity in retinal cells, classifications of other VUSs in RPGR and other IRD-causing genes can be expedited.

## 5. Conclusions

We have identified a novel *RPGR* variant that causes abnormal splicing, resulting in nonsense-mediated decay of the *RPGR* mRNA, with residual transcripts that escape nonsense-mediated decay likely subject to early truncation of the RPGR protein in patient fibroblasts, iPSC-RPE and iPSC-ROs. Notable features on phenotypic examination of patient iPSC-RPE and iPSC-ROs included reduced RPGR protein in the TZ of the primary and photoreceptor cilium, respectively. In the patient iPSC-RO photoreceptor cells, this led to a lack of co-localization with CEP290 in the region of the ciliary TZ, which would be expected to cause dysfunction in anterograde transport along the cilium. Other phenotypic features in patient iPSC-ROs included displaced rhodopsin and L/M opsin staining, increased photoreceptor apoptosis and increased F-actin staining. The presence of abnormal splicing induced by the *RPGR* c.1419-9A>G variant and abnormal staining patterns and phenotypic features in iPSC-RPE and iPSC-ROs facilitated reclassification of the variant from a VUS to a pathogenic variant. Identification of appropriate markers for the rapid and appropriate classification of VUSs in *RPGR* and other IRD-causing genes is necessary for informed patient genetic information and effective future therapeutic avenues.

## Figures and Tables

**Figure 1 jpm-12-00502-f001:**
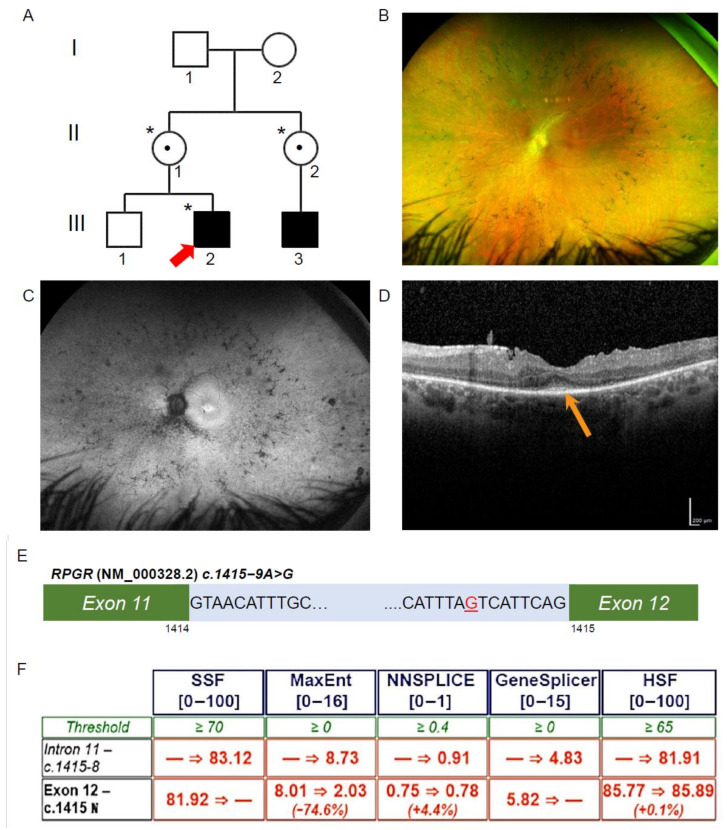
Ophthalmic investigations and *RPGR* novel variant. (**A**) The proband (Patient III-2; red arrow) was affected with RCD, as was his maternal cousin (III-3). The proband was hemizygous for the intronic variant in RPGR (*), and his mother and maternal aunt were heterozygous for the same variant (*), while his unaffected brother did not have the variant. Known female carriers are denoted (•). (**B**) Patient III-2 fundal images demonstrating mild pigmentary disturbance in the mid periphery. (**C**) Ultra-widefield fundus autofluorescence demonstrating a narrow ring of hyperautofluorescence around the fovea and patchy hypoautofluorescence scattered throughout the fundus. (**D**) The Spectral Domain Optical Coherence Tomography demonstrated a residual blurred ellipsoid zone at the fovea (orange arrow). (**E**) Patient III-2, *RPGR*, c.1415 − 9A>G highlighted in red in intron 11 of *RPGR* (NM_000328.2). (**F**) In silico prediction across five programs using Alamut Visual scored above the threshold for the novel variant c.1415 − 9G>A behaving as an acceptor site. Scores were below threshold for the canonical acceptor site, indicating a potential loss of function in the presence of the novel variant.

**Figure 2 jpm-12-00502-f002:**
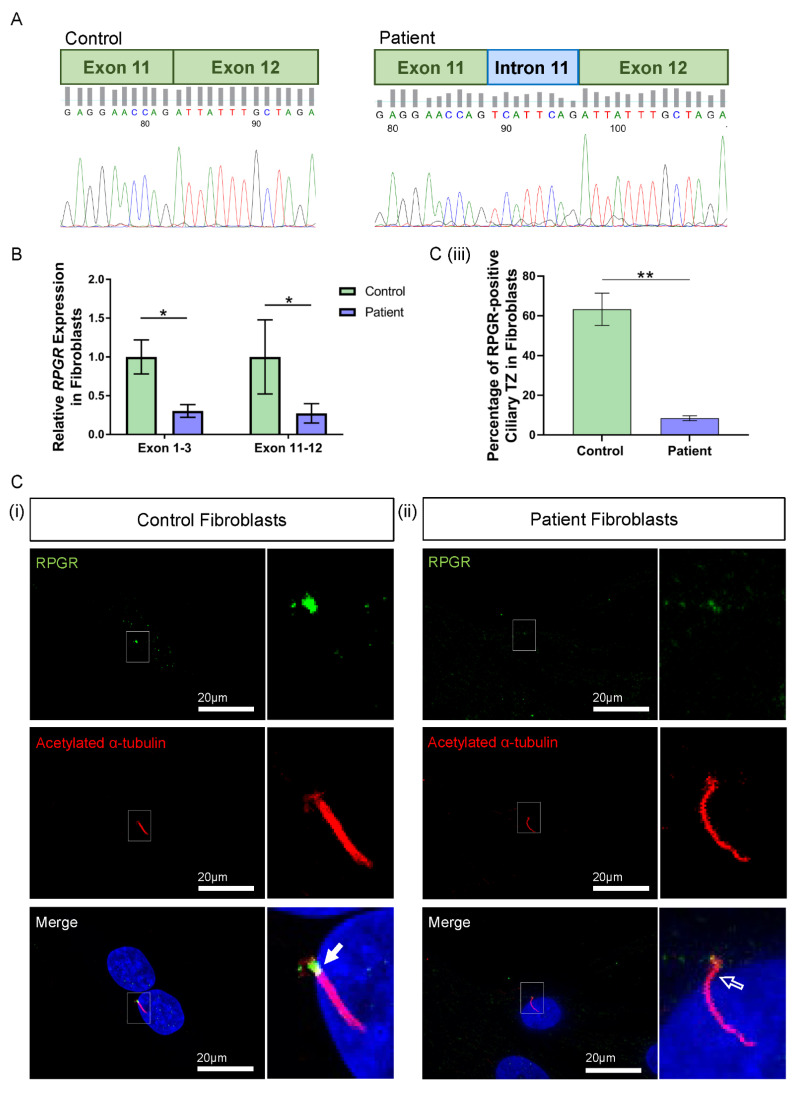
RNA and protein studies in *RPGR* c.1415 − 9A>G variant fibroblast cells. (**A**) Sanger sequencing of PCR-amplified cDNA across the exon 11–12 junction showed an additional eight base pairs (blue intron) interrupting canonical splicing (green exons) in the patient cells, compared to the control. (**B**) The expression of *RPGR,* determined by RT-qPCR, was decreased in patient fibroblasts compared to Control 1 cells at exons 1–3 and exons 11–12 (unpaired t-test, * *p* < 0.05, SEM from *n* = 3 independent experiments per line). Expression levels are relative to both the *HPRT* and *POLR2A* housekeeper gene expression. (**C**) (i) Immunofluorescence staining of the primary cilium in control fibroblast cells using acetylated α-tubulin (red) showed RPGR (green) localisation to the ciliary TZ (white arrow). (ii) The TZ localization of RPGR was lost in the patient’s fibroblast cell cilia (white arrow outline). (iii) Quantification of RPGR presence at the cilia TZ confirmed a decreased presence of RPGR at the TZ in the patient’s cells compared with the control (unpaired t-test, ** *p* < 0.01, SEM from *n* = 3 independent experiments per line, *n* = 159 control cilia counted, *n* = 219 patient cilia counted). The number of cilia with RPGR staining in the ciliary TZ was manually counted, and this was expressed as a percentage of the total number of cilia present.

**Figure 3 jpm-12-00502-f003:**
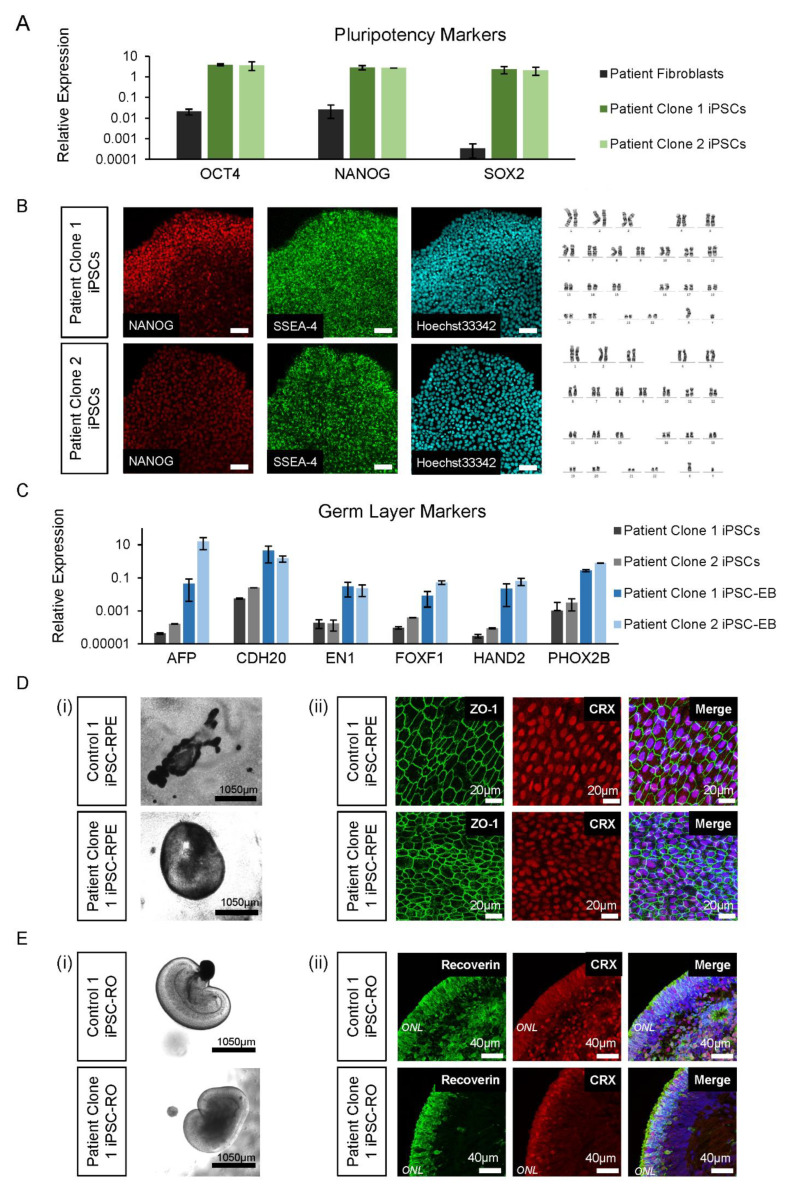
iPSC and differentiated retinal cell characterisation. (**A**) The patient’s iPSC expression of pluripotency markers *OCT4*, *NANOG* and *SOX2* was higher compared to fibroblast cells. Error bars = SEM. (**B**) The patient’s iPSC lines expressed NANOG and SSEA-4, with normal Hoechst33342 staining and karyotype (46, XY). (**C**) Expression of germ layer markers of the endoderm (*AFP*, *CDH20*), ectoderm (*EN1* and *PHOX2B*) and mesoderm (*FOXF1* and *HAND2*) were increased in iPSC-EBs compared to iPSCs. Error bars = SEM. (**D**) Control and patient pigmented iPSC-RPE islands (i) were isolated to allow growth into their classical hexagonal shape, revealed by ZO-1 staining (ii) and CRX expression. (**E**) Early control and patient iPSC-ROs at 6 weeks developed striations (i) and by 17 weeks, formed photoreceptor cells marked by the expression of recoverin and CRX (ii), which are early markers of photoreceptor development.

**Figure 4 jpm-12-00502-f004:**
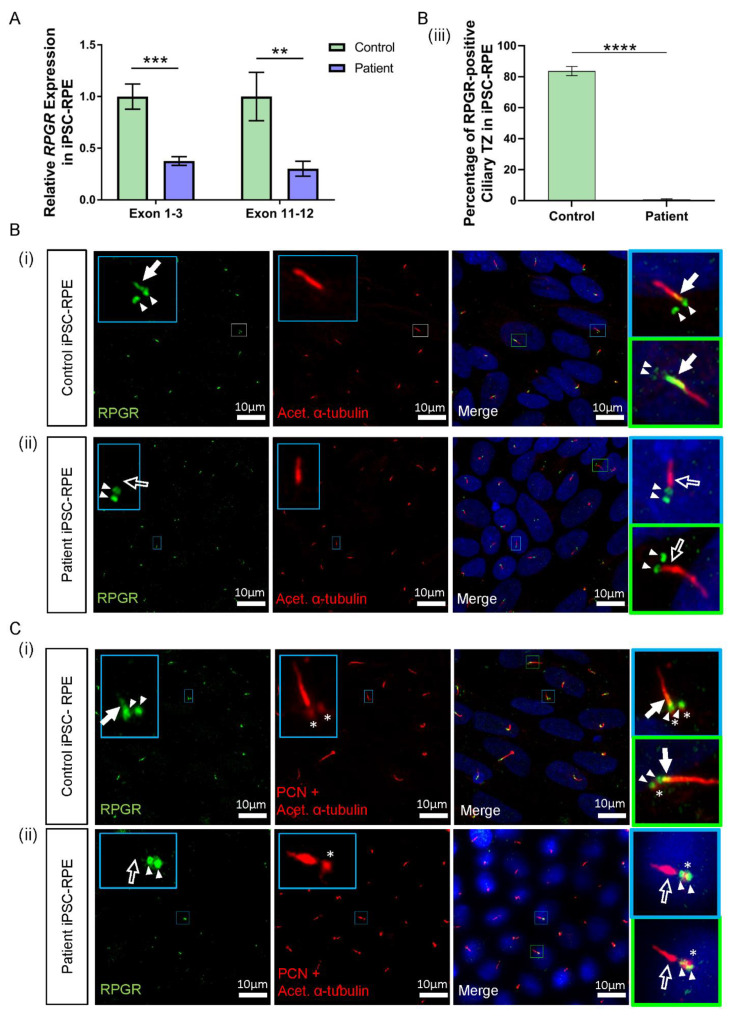
RNA and protein studies in *RPGR* c.1415 − 9A>G variant iPSC-RPE cells. (**A**) The expression of *RPGR* determined by RT-qPCR was decreased in patient iPSC-RPE compared to control cells at exons 1–3 and exons 11–12 (unpaired t-test, ** *p* < 0.01, *** *p* < 0.001; SEM error bars from *n* = 4 independent experiments per line). Expression levels are relative to both *HPRT* and *POLR2A* housekeeper genes. (Controls = Controls 1 and 2; Patient = Clones 1 and 2). (**B**) (i) Control iPSC-RPE cells were stained with acetylated α-tubulin (red), which stained the primary cilium. Staining with RPGR (green) showed the presence of RPGR in the TZ region of the cilium (white arrow), with this region showing as yellow on the merged image (white arrow). Adjacent RPGR punctate staining was also present (paired white arrowheads). (Control = Control 1). (ii) Patient iPSC-RPE cells demonstrated a striking lack of RPGR staining in the cilium TZ region (white arrow outline), with residual adjacent punctate staining (paired white arrowheads). (Patient = Patient Clone 1). These findings are also demonstrated in (**C**) (i) and (ii). (iii) Quantification of the RPGR presence at the ciliary TZ confirmed decreased levels in patient cells compared with control lines. Values are expressed as a percentage of the total number of cilia present (unpaired *t*-test, **** *p* < 0.0001, SEM from *n* = 4 independent experiments per line, *n* = 159 control cilia counted, *n* = 219 patient cilia counted). (Control = Controls 1 and 2; Patient = Clones 1 and 2). (**C**) (i) In Control iPSC-RPE cells, localisation of the RPGR punctate staining (paired white arrowheads) was determined to be at the basal bodies by co-staining with PCN (red, white asterisks). (Control = Control 2). (ii) Patient iPSC-RPE cells also showed a co-localisation of RPGR staining with the PCN staining. (Patient = Clone 2).

**Figure 5 jpm-12-00502-f005:**
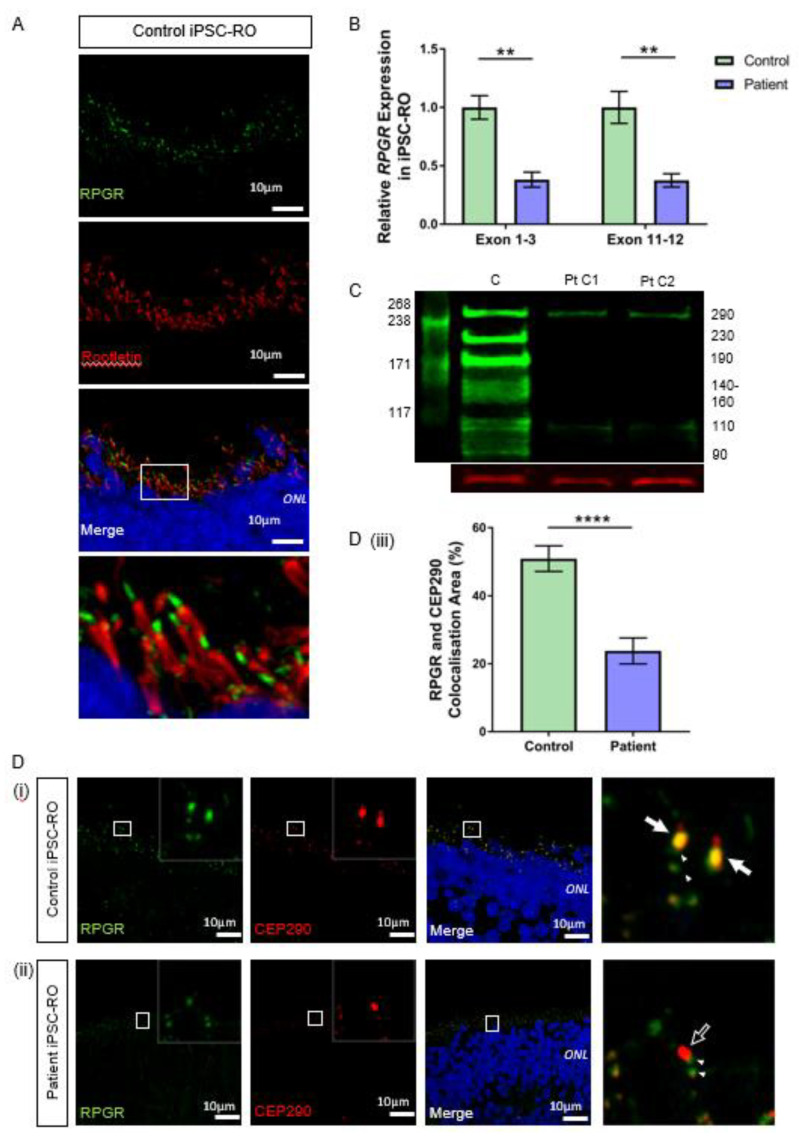
RNA and protein studies in *RPGR* c.1415 − 9A>G variant iPSC-ROs. (**A**) Normal localisation of RPGR was shown to be distal to rootletin staining. Rootletin marks the base of the primary cilia, so in the photoreceptor cells of the retinal iPSC-ROs, RPGR is localising to the transitional zone of the cilium. (**B**) The expression of *RPGR* determined by RT-qPCR was decreased in patient iPSC-ROs (30 weeks) compared to control cells at exon 1-3 and exon 11-12 (unpaired t-test, ** *p* < 0.01; SEM from *n* = 4 independent experiments, mRNA extracted from *n* = 5 pooled iPSC-ROs per experiment per line). Expression levels are relative to both *HPRT* and *POLR2A* housekeeper genes. (Control = Controls 1 and 2; Patient = Clones 1 and 2). (**C**) A western blot analysis of control (C) iPSC-ROs revealed multiple bands typical of RPGR (green). Patient (Pt C1; Pt C2) iPSC-ROs showed a reduction in 290 kDa and 110 kDa bands, with a loss of bands at 230 kDa, 190 kDa, 140–160 kDa, 125 kDa and 90 kDa. Vinculin (red) was used as a loading control. (Control = Controls 1 and 2; Patient = Clones 1 and 2; *n* = 4 individual experiments). (**D**) Control iPSC-RO and patient iPSC-RO photoreceptor cells at an age of 30 weeks were stained with antibodies against RPGR (green) and CEP290 (red), which localizes mainly to the TZ of photoreceptor cilia. (i) Control iPSC-ROs showed strong yellow co-staining (white arrow) along the ciliary TZ (Control = Control 1). (ii) Patient iPSC-ROs showed a lack of RPGR staining and hence a lack of co-staining with CEP290 at the photoreceptor TZ (white arrow outline). In both the control and patient cells, there was a presence of RPGR staining just away from the TZ region, which may be consistent with the localization in the basal bodies (green punctate staining denoted by white arrowheads). (Patient = Clone 2). (iii) An image analysis using ImageJ verified that control photoreceptor cilia have a larger overlapping area in the yellow co-staining of CEP290 with RPGR, whereas in patient cilia, this is lost (unpaired t-test, **** *p* < 0.0001; SEM from *n* = 4 independent iPSC-ROs per line). The co-localisation area was normalised against the CEP290 area and expressed as a percentage. (Control = Control 1 and 2; Patient = Clones 1 and 2).

**Figure 6 jpm-12-00502-f006:**
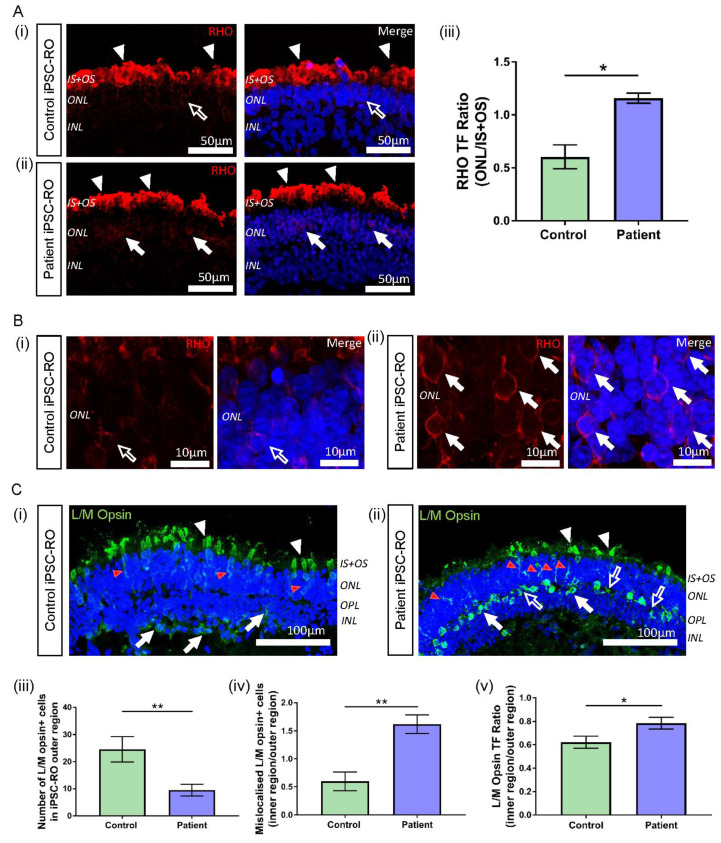
Rhodopsin and L/M opsin abnormalities in *RPGR* c.1415 − 9A>G iPSC-ROs. (**A**) (i) Rhodopsin staining (red) in control iPSC-ROs showed protein localization to the IS + OS region of the rod photoreceptor cells (white arrowheads), with minimal staining seen in the soma of the cells (white arrow outline). (ii) In patient iPSC-ROs, rhodopsin staining was also seen in the IS + OS region of the rod photoreceptor cells (white arrowheads), as well as in the soma (white arrows). (Control = Control 1, Patient = Clone 1). (iii) TF expressed as a ratio (ONL/IS + OS) indicated more staining present in the soma of rod photoreceptor cells in the ONL region in patient iPSC-ROs compared with controls (unpaired t-test, * *p* < 0.05, SEM from *n* = 3 independent iPSC-ROs per line). (Control = Control 1; Patient = Clones 1 and 2). (**B**) High resolution 63x Airyscan images of the ONL in (**i**) control iPSC-ROs demonstrated minimal rhodopsin staining at the cell soma (white arrow outline) compared to (ii) patient iPSC-ROs, where rhodopsin staining was more evident around the cell soma (white arrows). (Control = Control 1, Patient = Clone 1). (**C**) (i) The expression of L/M opsin (green) was seen in the control iPSC-RO photoreceptor IS + OS (white arrowheads) and ONL (red arrowheads), with some staining in the OPL and INL (white arrows). (ii) This pattern of staining was also seen in patient iPSC-RO IS + OS (white arrowheads) and ONL (red arrowheads), with L/M opsin staining also seen in the OPL and INL (white arrows). Additional L/M opsin staining was seen in the OPL that was outside the soma staining (white arrow outlines). (Control = Control 2; Patient = Clone 2). (iii) The quantification of L/M opsin+ cells in the outer region (IS + OS and ONL) of control and patient iPSC-ROs showed a decrease in cone photoreceptor cells in the patient iPSC-ROs (unpaired t-test, ** *p* < 0.01, SEM from *n* = 3–5 independent iPSC-ROs per line). (Control = Controls 1 and 2; Patient = Clones 1 and 2). (iv) The number of L/M opsin+ cone photoreceptor cells in the outer region (IS + OS and ONL) of control and patient iPSC-ROs compared against the number of L/M opsin+ cone photoreceptors mislocalised to the inner regions (OPL and INL) of the iPSC-ROs indicated more mislocalised cone photoreceptor cells in the patient iPSC-ROs (unpaired t-test, ** *p* < 0.01, SEM from *n* = 3–5 independent iPSC-ROs). (v) TF expressed as a ratio of inner/outer (INL + OPL/IS + OS + ONL) regions indicated more L/M opsin protein present in the OPL and INL of patient iPSC-ROs compared to controls (unpaired t-test, * *p* < 0.05, SEM from *n* = 3–5 independent iPSC-ROs per line). (Control = Controls 1 and 2; Patient = Clones 1 and 2).

**Figure 7 jpm-12-00502-f007:**
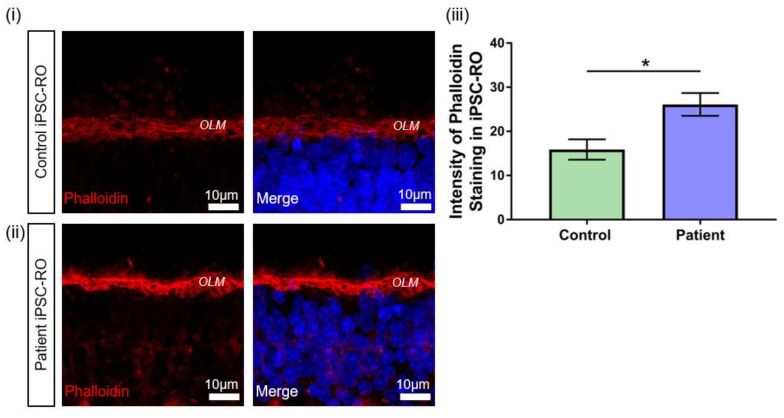
F-actin abnormalities in *RPGR* c.1415 − 9A>G iPSC-ROs. (**i**) Phalloidin expression in a 30-week-old (**i**) control and (**ii**) patient iPSC-ROs showed high intensity staining of phalloidin present in the control and patient iPSC-ROs at the OLM. (Control = Control 1, Patient = Clone 2). (**iii**) Pixel intensity showed that phalloidin staining was more intense in the patient iPSC-ROs, indicating an overexpression of F-actin in patient iPSC-ROs (paired t-test, * *p* < 0.05, SEM from *n* = 4 independent iPSC-ROs per line) compared to control iPSC-ROs.

## Data Availability

Novel genomic variant details will be available in the ClinVar database (Accession number SCV002095592). Appendix A from this study are available through contact with the corresponding author.

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
