# Peer review of "Human iPSC-Derived Retinal Organoids and Retinal Pigment Epithelium for Novel Intronic RPGR Variant Assessment for Therapy Suitability"

_jpm, 2022, doi:10.3390/jpm12030502_

Round 1

Reviewer 1 Report

In general, this is a well-defined study with clear rationales and experimental procedures. Scientific interpretations of the results are reasonable and sound. The data from the current study provided a great value to fill in knowledge gaps, given the fact that RPGR gene mutations are highly relevant to X-linked retinitis pigmentosa. The manuscript was well-written while the images with high resolution should be provided. The reviewer appreciated the thorough discussion, especially on the limitation of this study. Minor comments are listed below.

  1. Section 2: Great details in patient's ophthalmic medical history. Noting that some patient's information are spreading out throughout the text, it would be helpful to provide these information (gender, age and race, etc) in this section as well.
  2. Line 122: Please provide information or cite the reference article for isolating/developing primary fibroblast cell line.
  3. Line 212: Suggest making the assessment using Alamut Visual Software a separate section (potential title: in silico approach or prediction) with more detailed description such as criteria or parameters for analysis to help explain the contents in Figure 1F.
  4. 1A: Suggest labeling numbers to each individual to help readers easily identify II-1, III-2, etc
  5. The Y axis title in Fig.2 Ciii is confusing. It is not clear whether the plot represents the intensity or other measurement. Since the legend indicated that it is calculated as dividing average number of RPGR present at ciliary TZ by total number of ciliary, it is suggested changing the title to "percentage of RPGR-positive ciliary TZ". Same for Fig.4Biii.
  6. Fig.5C: The bottom bands for Vinculin are very faint. Please replace.
  7. Have the authors submitted the mutation characterized in the current study to any of gene mutation database? Such as https://databases.lovd.nl/shared/genes/RPGR (PMID: 18361418)?

Author Response

Reviewer #1 Comments to the Author:

Comment 1:

Section 2: Great details in patient's ophthalmic medical history. Noting that some patient's information are spreading out throughout the text, it would be helpful to provide these information (gender, age and race, etc) in this section as well.

Response: These details have been included in section 2 under the heading “Patient ophthalmic investigations” and reads “Ophthalmic investigations were undertaken in a 34-year-old Caucasian male (proband) and included visual acuity assessment, ultra-widefield pseudocolour fundus photos and autofluorescence (Optos plc, Dunfermline, UK)” (line 106).

Comment 2:

Line 122: Please provide information or cite the reference article for isolating/developing primary fibroblast cell line.

Response: Appropriate reference has been included (lines 140-141).

Comment 3:

Line 212: Suggest making the assessment using Alamut Visual Software a separate section (potential title: in silico approach or prediction) with more detailed description such as criteria or parameters for analysis to help explain the contents in Figure 1F.

Response: Reference has been made to our previous publication where methodology for Alamut Visual Software is described in detail. Additionally, a description of the process from variant identification to the use of the splice prediction software has been added. This is in Section 2. The heading has now been changed to include in silico splice prediction, and now reads as: “Exome sequencing, analysis and in silico splice prediction” (lines 111), with the following information added: “Subsequent reads were aligned to the hg19 reference genome and further filtered by examination against a gene panel of 68 known RCD disease genes. This analysis identified a novel variant in RPGR: c.1415-9A>G. In silico pathogenicity analysis was undertaken on the novel variant using Alamut Visual Software (Version 2.5, Interacive Biosoftware, France), as per our previous studies [17]. The variant was assessed using the Alamut Visual splicing prediction tool, which amalgamated algorithmic data from five distinct splice prediction programs (SSF, MaxEnt, NNSPLICE, GeneSplicer, and HSF) where consensus across the algorithms was required for the variant to be considered significant” (lines 115-121)

Comment 4:

1A: Suggest labeling numbers to each individual to help readers easily identify II-1, III-2, etc

Response: Individuals in the pedigree have now been labelled with the appropriate number for clarity of identification.

Comment 5:

The Y axis title in Fig.2 Ciii is confusing. It is not clear whether the plot represents the intensity or other measurement. Since the legend indicated that it is calculated as dividing average number of RPGR present at ciliary TZ by total number of ciliary, it is suggested changing the title to "percentage of RPGR-positive ciliary TZ". The same was done for Fig.4Biii.

Response: The Y-axis labels have been updated to better reflect the data presented.

Comment 6:

Fig.5C: The bottom bands for Vinculin are very faint. Please replace.

Response: Vinculin bands have been replaced.

Comment 7:

Have the authors submitted the mutation characterized in the current study to any of gene mutation database? Such as https://databases.lovd.nl/shared/genes/RPGR (PMID: 18361418)?

Response: The mutation has been submitted to ClinVar, with accession number SCV002095592. This is referenced in the manuscript under the Data Availability Statement and reads “Novel genomic variant details will be available in the ClinVar database (Accession number SCV002095592)” (line 635).

Reviewer 2 Report

In this manuscript, the authors described and validated the clinical significance of an intronic RPGR variant, c.1415-9A>G, contributing to X-linked rod-cone dystrophy (RCD). This variant previously was the variant of unknown significance. This is a very interesting and clinically significant work since it contributes towards the importance of this variance for the prognosis and diagnosis of rod-cone dystrophy. RGD is responsible for severe vision impairment. Please include a couple of points as follows to detail the article further:

1. In the introduction, mention and discuss the already known clinically significant variants for RGD and how they differ from  c.1415-9A>G in RPGR when it comes to the disease manifestation.

2. At the moment, the technical details about the exome and sanger's sequencing are hardly mentioned in the materials and methods. Things like the depth of sequencing, reference genome used for comparison, filtering methods employed, etc should be mentioned. 

Author Response

Reviewer #2 Comments to the Author:

Comment 1:

In the introduction, mention and discuss the already known clinically significant variants for RGD and how they differ from c.1415-9A>G in RPGR when it comes to the disease manifestation.

Response: Literature background relating to genotype-phenotype associations in RPGR-associated disease has been added into the introduction and reads: “RPGR-associated disease can be caused by missense, in-frame insertion/deletion, frameshift and splice site variants located throughout exons 1-14 and can lead to nonsense-mediated decay and loss of function of the RPGR protein. In addition, insertion and deletion variants in the mutational hotspot region, ORF15, may result in protein truncation and more severe clinical presentations [4, 5]. Approximately 70% of pathogenic RPGR variants result in predominately rod photoreceptor disease, while cone or cone-rod dystrophies are found in the remainder [6, 7]. Variants in exon 1-14 and the proximal part of ORF15 are more frequently associated with rod-dominated disease, while those in the more 3’ region of ORF15 are more usually associated with a cone or cone-rod phenotype [6]. Splice variants generally reflect this pattern, with studies indicating correct splicing of both isoforms is integral for normal RPGR function [8-10] (lines 67-75)

The following section has been added in the discussion, paragraph 2, to indicate the relationship of the phenotype to the variant investigated in this study: “…with a predominance of the RCD phenotype in the exon 1-14 region [6, 7]. The RCD phenotype in the patient in this study is consistent with variant localisation in the exon 1-14 region of RPGR” (lines 513-514).

Comment 2:

At the moment, the technical details about the exome and sanger's sequencing are hardly mentioned in the materials and methods. Things like the depth of sequencing, reference genome used for comparison, filtering methods employed, etc should be mentioned. 

Response: Depth of sequencing, reference genome and filtering methods have now been included under the heading “Exome sequencing, analysis and in silico prediction”, and reads: “TruSight One Clinical Exome analysis has an average depth of coverage at 160x, and is filtered to exclude regions with <15x read depth. Allele frequency cut-off was >0.01. Subsequent reads were aligned to the hg19 reference genome and further filtered by examination against a gene panel of 68 known RCD disease genes. This analysis identified a novel variant in RPGR: c.1415-9A>G.” (lines 114-117). Sanger sequencing was done by AGRF in Sydney, and further details have been added: “Variant confirmation and segregation studies were performed on PCR amplicons, by bi-directional Sanger sequencing on an ABI3730xl instrument, undertaken at the Australian Genome Research Facility (Westmead, Australia)” (lines 121-122).

Reviewer 3 Report

1] Protein expression of retinal organoids from patients vs. controls should be repeated. RPGR and other IFT (to test for function in trafficking) antibodies should be used for analysis. The RPGR Ab from sigma has not been validated for WB analysis.

2] Fig 6A, Rhodopsin mislocalization to ONL is not convincing as stated by the authors. What is the source of the Rho Ab, I could not find this information in the Methods.

3] Fig 7, F-actin staining should be repeated. The image provided has quality issues.

4] Was apoptosis evaluated in these organoids? if authors claim mislocalization of key OS proteins, then this should lead to cellular apoptosis. Please use TUNEL assay.

5] H&E staining should also be performed on these control and patient organoids

Round 2

Reviewer 3 Report

1] Authors claim in their abstract, results, discussion, and elsewhere that Mislocalisation of rhodopsin staining was present in patient iPSC-RO rod photoreceptor cells. The results/images shown are not convincing.

As per their abstract, variants in RPGR cause RCD, however, in their organoids, cone opsin mislocalization is very much apparent, than what is observed in rods.

Authors further claim in their rebuttal that they now observed a "subtle increase in mislocalization of rhodospin"...how can this be quantified and concluded as being significant?

2] The Sigma Rho antibody used in their study has not been previously validated in any previous study. I would like to see the 1d4 antibody (Sigma/Abcam) being used.

3] Their response to my earlier concerns (see comment 1) was not addressed. A mere comparison between antibodies is not validation. Optimization is not validation. Positive and negative controls need to be included, when Ab's are being used for the first time and where there is no literature supporting the specificity.

At the minimum wild-type mouse and normal human retinal lysates should be tested for RPGR Ab specificity. Mass spec could also confirm their WB data.

4] Figure 7, same issues as previously raised still exist in their revised manuscript. In control iPSC, the nuclei/DAPI staining is brighter than those in the Patient iPSC. This raises concern that laser intensities were not equal/consistence during the capture of these images.

Images should be re-taken at the same intensities.

5] please check if scale bars are correct in Suppl. Fig S5A ii; Nuclei/DAPI in patients iPSC appear smaller than the images in Suppl. Fig S5A i (controls).
